# ZSON: Zero-Shot Object-Goal Navigation using Multimodal Goal Embeddings

**Arjun Majumdar,**\* **Gunjan Aggarwal,**\* **Bhavika Devnani,** **Judy Hoffman,** **Dhruv Batra**

Georgia Institute of Technology

## Abstract

We present a scalable approach for learning *open-world* object-goal navigation (`ObjectNav`) – the task of asking a virtual robot (agent) to find any instance of an object in an unexplored environment (e.g., *"find a sink"*). Our approach is entirely *zero-shot* – i.e., it does not require `ObjectNav` rewards or demonstrations of any kind. Instead, we train on the image-goal navigation (`ImageNav`) task, in which agents find the location where a picture (i.e., goal image) was captured. Specifically, we encode goal images into a multimodal, semantic embedding space to enable training semantic-goal navigation (`SemanticNav`) agents at scale in unannotated 3D environments (e.g., HM3D). After training, `SemanticNav` agents can be instructed to find objects described in free-form natural language (e.g., *"sink," "bathroom sink,"* etc.) by projecting language goals into the same multimodal, semantic embedding space. As a result, our approach enables open-world `ObjectNav`. We extensively evaluate our agents on three `ObjectNav` datasets (Gibson, HM3D, and MP3D) and observe absolute improvements in success of 4.2% - 20.0% over existing zero-shot methods. For reference, these gains are similar or better than the 5% improvement in success between the Habitat 2020 and 2021 `ObjectNav` challenge winners. In an open-world setting, we discover that our agents can generalize to compound instructions with a room explicitly mentioned (e.g., *"Find a kitchen sink"*) and when the target room can be inferred (e.g., *"Find a sink and a stove"*).

## 1  Introduction

Imagine asking a home assistant robot to find a *"flat-head screwdriver"* or the *"medicine case near the bathroom sink."* Building such assistive agents is a problem of deep scientific and societal value.

To study this problem systematically, the embodied AI community has rallied around a problem called object-goal navigation ( `ObjectNav`) [1]. Given the name of an object (e.g., *"chair"*), `ObjectNav` involves exploring a 3D environment to find any instance of the object. The last few years have witnessed the development of new environments [2–6], annotated 3D scans [7–9], datasets of human demonstrations [10], and approaches for ObjectNav [11–16], cumulatively leading to strong progress. For instance, the entries in the annual Habitat challenge [17] have jumped from 6% success (DD-PPO baseline in 2020) to 53% success (top entry in ongoing 2022 Habitat Challenge public leaderboard).

While this progress is exciting, we believe that a subtle but insidious assumption has snuck into this line of work: the closed-world assumption. We started by discussing an open-world scenario where a person may describe any object in language (e.g., *"flat-head screwdriver"*), but `ObjectNav` is currently formulated over a closed predetermined vocabulary of object categories (*"chair"*, *"bed"*,

---

\*equal contribution

36th Conference on Neural Information Processing Systems (NeurIPS 2022).

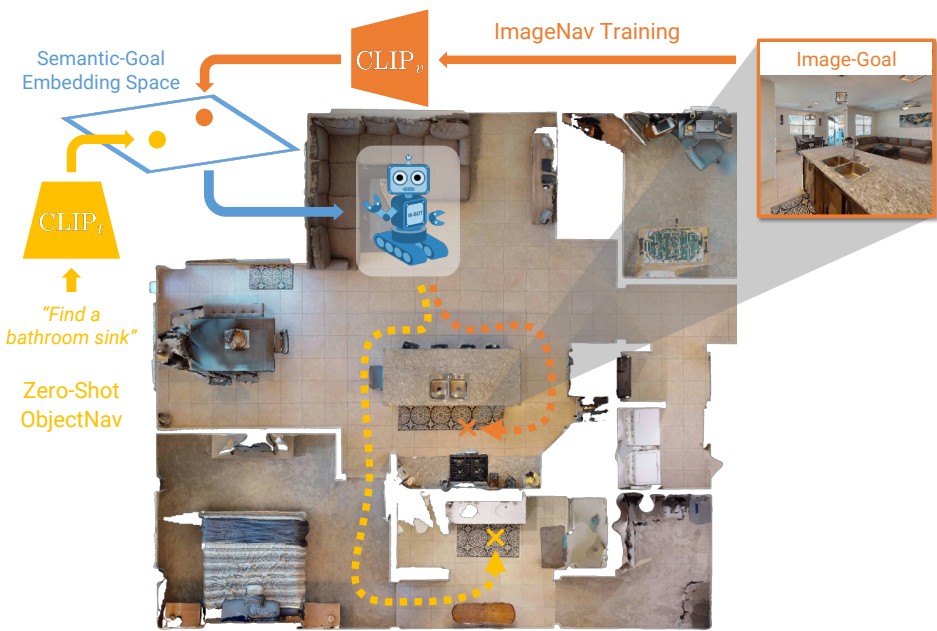

Figure 1: We propose projecting navigation goals (from images or text) into a common, semantic embedding space using a pre-trained vision and language model (CLIP). This allows agents trained with image-goals to understand goals expressed in free-form natural language (e.g., *"Find a bathroom sink."*). Accordingly, our approach enables *open-world* object-goal navigation in a *zero-shot* manner – i.e., without using ObjectNav rewards or demonstrations for training.

*"sofa"*, etc.), with approaches using pre-trained object detectors and segmenters for these categories [10–13]. While this assumption may have been essential to get started on this problem, it is now important to move beyond it and ask – how can embodied agents find objects in an open-world setting?

In this work, we develop an approach for `ObjectNav` that is both *zero-shot*, i.e., does not require *any* `ObjectNav` rewards or demonstrations, and *open-world*, i.e., does not require committing to a taxonomy of categories. Our key insight is that we can create a visiolinguistic embedding space to decouple two problems – (1) describing and representing semantic goals (*"chair"*, *"brown chair"*, picture of brown chair) from (2) learning to navigate to semantic goals.[2]

To represent semantic goals (1), we leverage recent advances in multimodal AI research on learning a common embedding space for images and text using large collections of image-captions pairs. Specifically, we use CLIP [19], a method for training dual vision and language encoders that produce similar representations for paired data such as an image and its caption. As shown in Fig. 1, we use CLIP to transform image-goals (e.g., a picture of the kitchen island) and object-goals (e.g., *"bathroom sink"*) into *semantic-goals* representing navigation targets. Our main observation is that a semantic-goal produced from an image (e.g., a picture of the bathroom sink) should be similar to semantic goals produced from descriptions of the same target (e.g, *"bathroom sink"*). Thus, we hypothesize that these modalities (images and language) can be used interchangeably for creating semantic goals.

Accordingly, for learning to navigate to semantic goals (2), we train agents using image-goals encoded via CLIP's image encoder. Then, we evaluate the learned navigation policy on `ObjectNav`, where goals are specified in language (e.g., *"chair"*) and encoded via CLIP's text encoder. As a result, our agents perform `ObjectNav` without ever directly training for the task – i.e., in a zero-shot manner.

An important advantage of our approach is that it reduces the data labeling burden. Image-goals can be procedurally generated by randomly sampling points in 3D environments. This is in stark contrast to `ObjectNav`, which requires annotating 3D meshes [7–9] and potentially collecting large-scale human demonstrations [10] for training. Secondly, the interface to our agents is a natural language description – matching the grand vision that inspired the `ObjectNav` task. Through this interface we can refine

---

[2]Similar arguments have been made by Al-Halah et al. [18]. A detailed discussion is provided in Section 2.

object-goals by, for instance, specifying object attributes (*"brown chair"*) or indicating which room the object is in (*"bathroom sink"*) – which is not possible with traditional `ObjectNav` agents.

We perform large-scale experiments on three `ObjectNav` datasets – Gibson [4], MP3D [8], and HM3D [20]. Our zero-shot agent (that has not seen a single 3D semantic annotation or `ObjectNav` training episode) achieves a 31.3% success in Gibson environments, which is a 20.0% absolute improvement over previous zero-shot results [18]. In MP3D, our agent achieves 15.3% success, a 4.2% absolute gain over existing zero-shot methods[21]. For reference, these gains are on par or better than the 5% improvement in success between the Habitat 2020 and 2021 `ObjectNav` challenge winners. On HM3D, our agent's zero-shot SPL matches a state-of-the-art `ObjectNav` method [16] that trains with direct supervision from 40k human demonstrations.

Additionally, we study two techniques that are used in our approach to improve zero-shot `ObjectNav` performance. First, we find that pretraining the visual observation encoder has an outsized effect on zero-shot transfer. Specifically, success on the `ImageNav` training task improves 4.5% - 5.8%, while downstream success on zero-shot `ObjectNav` improves by 9.4% - 10.4%. Similarly, increasing the number of training environments (from 72 to 800) leads to a small drop in `ImageNav` success, but results in a substantial improvement of 6.6% in success on zero-shot `ObjectNav`.

Finally, we qualitatively experiment with an open-world setting and observe that our `SemanticNav` agents can properly change behavior in response to instructions that include room information. For instance, when finding a *"bathroom sink"* the agent does not enter the kitchen, and when looking for a *"kitchen sink"* it does not enter bathrooms. Furthermore, we observe similar room awareness patterns for instructions such as *"Find a sink and a stove,"* where the target room (*"kitchen"*) can be inferred. Source code for reproducing our results will be publicly released.

## 2 Related Work

Our work builds on research studying image-text alignment techniques (e.g., CLIP [19]) and their use in visual navigation. In this section, we discuss methods most related to our proposed approach.

**Image-Text Alignment Models.** Recent progress in vision-and-language pretraining has led to models such as CLIP [19], ALIGN [22], and BASIC [23] that can perform open-world image classification, and achieve strong performance on standard computer vision benchmarks (e.g., ImageNet [24]). These models learn visual representations by training on massive datasets of image-caption pairs scraped from the web (e.g., the 400M pairs used for CLIP or 6.6B for BASIC). In this work, we take advantage of the semantic representations learned by CLIP to project navigation goals (e.g., a picture of a brown chair or *"brown chair"*) into a multimodal, semantic-goal embedding space.

**CLIP for Visual Navigation.** A straightforward approach for using CLIP in a visual navigation agent is to process the agent's observations and navigation instructions (e.g., *"Find a chair"*) with the CLIP image and text encoders, then learn a navigation policy that operates on these embeddings. Such a solution was explored in EmbCLIP [25] with promising results. However, this approach requires ObjectNav rewards or demonstrations to supervise the navigation policy, which is difficult and costly to collect at scale. As a result, existing training datasets tend to be small and agents generalize poorly to new settings. For instance, EmbCLIP only achieves an 8% success rate in finding objects that were not used in training. By contrast, we train using the image-goal navigation task, which does not require annotated environments. Thus, we are able to scale training to 800 unannotated 3D scenes, which substantially improves generalization (as demonstrated in Section 5).

**Zero-Shot ObjectNav.** Two recent works [18, 21] directly address our motivation (zero-shot `ObjectNav`) and are most related. First, ZER [18] proposes a two-stage framework in which an image-goal navigation (`ImageNav`) agent is first trained from scratch. Then, independent encoders are trained to map from various modalities (including language) into the image-goal embedding space. A key challenge with this approach is that image-goal embeddings may not capture semantic information because semantic annotations are not used in `ImageNav` training. Instead, an `ImageNav` agent trained from scratch may learn to pattern match visual observations and goal image embeddings. By contrast, our approach reverses these two stages, with CLIP pretraining representing stage one. Thus, our approach uses a goal embedding space that captures semantics by design. We empirically demonstrate the benefits of our proposed approach in Section 5.

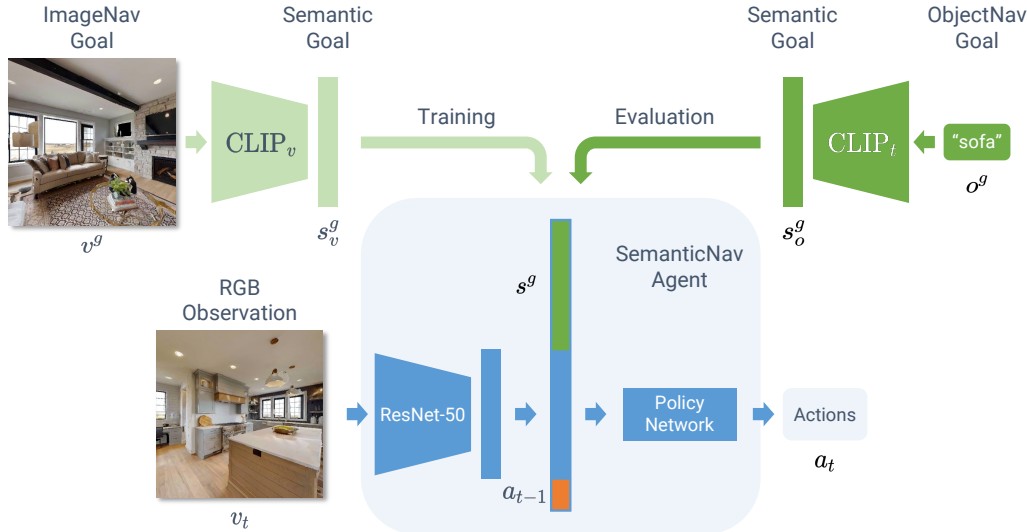

Figure 2: We tackle both `ImageNav` and `ObjectNav` via a common `SemanticNav` agent. This agent accepts a semantic goal embedding ($s^g$), which comes from either CLIP's visual encoder ($\text{CLIP}_v$) in ImageNav or CLIP's textual encoder ($\text{CLIP}_t$) in ObjectNav. Our agent has a simple architecture: RGB observations are encoded with a pretrained ResNet-50, and a recurrent policy network predicts actions using encodings of the goal $s^g$, observation, and the previous action $a_{t-1}$.

In concurrent work, CLIP-on-Wheels (CoW) [21] uses a gradient-based visualization technique (GradCAM [26]) with CLIP to localize objects in the agent's observations. This is combined with a heuristic exploration policy to enable zero-shot object-goal navigation. In contrast, we demonstrate that learning a navigation policy can substantially outperform the heuristic exploration approach proposed in [21] without using explicit object localization techniques.

# 3 Preliminaries: Image-Text Alignment and Image-Goal Navigation

**Image-Text Alignment Models.** Multimodal alignment models aim to learn a mapping from images $v$ and text $t$ into a shared embedding space such that representations for corresponding image-text pairs (e.g., a picture and its caption) are similar. Recent image-text alignment models [19, 22, 23] use a dual-encoder framework and optimize the InfoNCE [27] contrastive learning objective, which maximizes cosine similarity between representations of matching image-text pairs and minimizes similarity for non-matching pairs. In this work, we leverage CLIP [19], which was trained on 400M image-text pairs that cover a wide range of visual concepts.

**Image-Goal Navigation.** In image-goal navigation (`ImageNav`) [28], agents explore an environment to find the position where a goal-image $v^g$ was captured. We consider a setting in which both the goal-image and the agent's observations consist of RGB images taken from the agent's egocentric point of view. An agent can select from four actions: `MOVE_FORWARD` by 0.25m, `TURN_LEFT` by 30°, `TURN_RIGHT` by 30°, or `STOP`. The agent succeeds if it selects `STOP` within 1.0m of the goal.

An `ImageNav` episode is uniquely defined by a starting position and (reachable) goal viewpoint within a 3D environment. Thus, `ImageNav` training data can be procedurally generated without annotating the scene – i.e., the objects and rooms do not need to be labeled. As a result, the size of an `ImageNav` dataset is only limited by the number of environments available for training. In this work, we use `ImageNav` to train visual navigation agents at scale (in terms of the number of training environments).

# 4 Approach

This section describes our framework for training visual navigation agents. We use CLIP [19] to produce semantic goal embeddings of image-goals (e.g., a picture of the sink) and object-goals (e.g., *"sink"*). This allows training semantic-goal navigation agents at scale using image-goals in HM3D environments [20], then deploying these agents for object-goal navigation in a *zero-shot* manner. In other words, our agents execute object-goal navigation without ever directly training for the task.

## 4.1 Learning Semantic-Goal Navigation

As illustrated in Fig. 2 (top-left), given an image-goal $v^g$, we use a CLIP visual encoder $\texttt{CLIP}_v$ to generate a semantic goal embedding $s_v^g = \texttt{CLIP}_v(v^g)$ that is used to guide navigation. Conceptually, encoding image-goals with CLIP preserves semantic information about the goal, such as visual concepts that might be described in image captions (e.g., *"a sofa in a living room"*). However, semantic goal embeddings are less likely to include low-level features (e.g., the exact patterns in a wood floor) that do not correlate with web-scraped captions. While removing low-level information might make the navigation task more difficult, our goal is to learn a policy that transfers to $\texttt{ObjectNav}$ in which agents only receives high-level goals (e.g., *"Find a sofa"*). As an added benefit, generating semantic goal embeddings as a pre-processing step substantially improves training time (by ∼3.5x).

Our agent architecture is shown in Fig. 2. At each timestep $t$, our agent receives an egocentric RGB observation $v_t$ and a goal representation $s_v^g$. The observation is processed by a ResNet-50 [29] encoder, which is pretrained on the Omnidata Starter Dataset (OSD) [30] using self-supervised learning (DINO [31]) following the pretraining recipe presented in OVRL [16]. The output from the ResNet-50 encoder is concatenated with the goal representation $s_v^g$ and an embedding of the agent's previous action $a_{t-1}$ and then passed to the policy network composed of a two-layer LSTM. The policy network outputs a distribution over the action space.

We train our $\texttt{SemanticNav}$ agent with reinforcement learning (RL). During RL training, we use two data augmentation techniques: color jitter and random translation (adapted from [16]). Specifically, we train with DD-PPO [32] using a reward function proposed for $\texttt{ImageNav}$ by Al-Halah et al. [18]:

$$r_t = r_{\text{success}} + r_{\text{angle-success}} - \Delta_{\text{dtg}} - \Delta_{\text{atg}} + r_{\text{slack}} \tag{1}$$

where $r_{\text{success}} = 5$ if $\texttt{STOP}$ is called when the agent is within 1m of the goal position (and 0 otherwise), $r_{\text{angle-success}} = 5$ if $\texttt{STOP}$ is called when the agent is within 1m of the goal position and the agent is pointing within 25° of the goal heading – i.e., the direction the camera was pointing when the goal image was collected – (and 0 otherwise), $\Delta_{\text{dtg}}$ is the change in the agent's distance-to-goal – i.e., the geodesic distance to the goal position, $\Delta_{\text{atg}}$ is the change in the agent's angle-to-goal – i.e., the difference between the agent's heading and the goal heading – but is set to 0 if the agent is greater than 1m from the goal, and $r_{\text{slack}} = -0.01$ to encourage efficient navigation. In general, this reward function encourages both reaching the goal and looking towards the goal before calling $\texttt{STOP}$, which matches the requirements of the downstream $\texttt{ObjectNav}$ task.

## 4.2 Zero-Shot Object-Goal Navigation

Recall that in $\texttt{ObjectNav}$ [1], agents are given a target category (e.g., *"sofa"* or *"chair"*) and must locate any instance of that object (i.e., *"any sofa"* or *"any chair"*). Similar to $\texttt{ImageNav}$, $\texttt{ObjectNav}$ requires exploring new environments that the agent has never seen before. However, in $\texttt{ObjectNav}$, the goal (e.g., *"sofa"*) provides a minimal amount of information about where the agent must go and it requires recognizing any version of the goal object in the new scene.

To address this task, we transform object-goals $o^g$ (e.g., *"sofa"*) into semantic goal embeddings using the CLIP text encoder $\texttt{CLIP}_t$, which results in the semantic goal $s_o^g = \texttt{CLIP}_t(o^g)$. CLIP aligns image and text, thus the semantic goals from text $s_o^g$ should be close (in terms of cosine similarity) to the CLIP visual embeddings $s_v^g$ used in training. To keep our approach simple and easily reproducible, we do not use any prompt engineering (e.g., using a template such as "A photo of a <>"). Instead, we simply use the object name (e.g., *"sofa"*) as the object-goal input $o^g$.

# 5 Experimental Findings and Qualitative Results

This section studies the zero-shot `ObjectNav` performance of our proposed approach. First, we evaluate our method in the traditional `ObjectNav` setting [1] where agents must find any instance of the goal object (*"Find a chair"*). Then, we explore variations of `ObjectNav` in which additional information, such as a room location (e.g., *"bathroom sink"*), is given to refine the task. These experiments aim to demonstrate both the effectiveness and versatility of our approach.

## 5.1 Experimental Setup

**Training Dataset.** We generate a dataset for training our `SemanticNav` agent using the 800 training environments from HM3D [20]. First, we sample 9k `ImageNav` episodes for each HM3D scan, split equally between 3 difficulty levels corresponding with path length: EASY (1.5-3m), MEDIUM (3-5m), and HARD (5-10m). We follow the episode generation approach from [33]. This results in $9k \times 800 = 7.2M$ navigation episodes for training. Next, we pre-process the goal-images with the ResNet-50 version of CLIP [19] to produce 1024 dimensional semantic goal vectors $s_v^g$ for each navigation episode. During pre-processing, we further augment the dataset by sampling goal-images at four evenly-spaced heading angles to produce 36M total episodes for training. Sampling at multiple angles approximates the randomized sampling used in [18].

**Agent Configurations.** Two different agent configurations are frequently used in prior work on visual navigation. Configuration A is generally used for `ImageNav` and has an agent height of 1.5m, radius of 0.1m, and a single 128×128 RGB sensor with a 90° horizontal field-of-view (HFOV) placed 1.25m from the ground. Configuration B is typically used for `ObjectNav` and approximately matches a LoCoBot, with an agent height of 0.88m, radius of 0.18m, and a single 640×480 RGB sensor with a 79° HFOV placed 0.88m from the ground. Both configurations use the aforementioned step size of 0.25m and left and right turning angle of 30°.

**Evaluation Datasets.** We measure performance on one `ImageNav` and three `ObjectNav` datasets:

– `ImageNav` (Gibson) consists of 4,200 episodes from 14 Gibson [4] validation scenes. The dataset was produced by Mezghani et al. [33] for agents with configuration A.
– `ObjectNav` (Gibson) was generated by Al-Halah et al. [18] for agents with configuration A. The dataset consists of 1,000 episodes in 5 Gibson [4] validation scenes for 6 object categories.
– `ObjectNav` (HM3D), released with the Habitat 2022 challenge, consists of 2,000 episodes from 20 HM3D [20] validation scenes with objects from 6 categories, and uses agents with configuration B.
– `ObjectNav` (MP3D) released with the Habitat 2020 challenge, contains 2,195 episodes from 11 MP3D [8] validation scenes for 21 object categories, and requires agents with configuration B.

Due to the different agent configurations required by these evaluation datasets, we train agents with both settings to make fair comparisons with prior work on zero-shot `ObjectNav`. For all experiments, we report two standard metrics for visual navigation tasks: success rate (SR) and success rate weighted by normalized inverse path length (SPL) [34].

**Implementation Details.** We generate a `SemanticNav` dataset for each agent configuration (A and B). The CLIP ResNet-50 encoder processes $224 \times 224$ images. Accordingly, for configuration A, we render $512 \times 512$ RGB frames, then resize to $224 \times 224$. For configuration B, we render at $640 \times 480$, then resize and center crop. We train agents using PyTorch [35] and the Habitat simulator [2, 3]. Each training run was conducted on a single compute node with 8 NVIDIA A40 GPUs. We train agents for 500M steps, requiring ~1,704 GPU-hours to train two agents (one for each configuration). Additional training hyperparamters are detailed in the Appendix. We report results using the best checkpoint, selected based on `ObjectNav` validation success rate (SR). During evaluations we sample actions from the agent's output distribution. We report results averaged over three evaluation runs.

**Baselines.** We provide comparisons with the, to the best of our knowledge, only two existing zero-shot methods for object-goal navigation (`ObjectNav`):

– **Zero Experience Required (ZER)** [18]: first trains an `ImageNav` agent composed of two ResNet-9 encoders for processing the goal-image and agent observations, and a policy network consisting of

Table 1: **Zero-shot ObjectNav performance** on Gibson [4], HM3D [20], and MP3D [8] validation. All methods use a single RGB sensor for agent observations except CoW [21], which also uses depth observations and OVRL [16], which uses `GPS+Compass` for `ObjectNav`. Our approach (ZSON) substantially improves on previous zero-shot methods and narrows the gap to SOTA fully-supervised methods such as OVRL [16], which is not zero-shot and provided for reference. We report ZSON results averaged over three evaluation trials. The standard deviation in ZSON `ObjectNav` `SR` is 0.02% in Gibson, 0.46% in HM3D, and 0.11% in MP3D. *indicates reproduced results*

| Method | ImageNav (Gibson) | | ObjectNav (Gibson) | |
|---|---|---|---|---|
| | SPL | SR | SPL | SR |
| OVRL [16] | 27.0% | 54.2% | - | - |
| ZER [18] | 21.6% | 29.2% | - | 11.3% |
| ZSON (ours) | **28.0%** | **36.9%** | 12.0% | **31.3%** |

(a) Configuration `A`

| Method | ObjectNav (HM3D) | | ObjectNav (MP3D) | |
|---|---|---|---|---|
| | SPL | SR | SPL | SR |
| OVRL [16] | 12.3%* | 32.8%* | 7.0% | 25.3% |
| CoW [21] (w/depth) | - | - | **6.3%** | 11.1% |
| ZSON (ours) | 12.6% | 25.5% | 4.8% | **15.3%** |

(b) Configuration `B`

a 2-layer GRU. After training the navigation policy, a 2-layer MLP is trained to map from a goal object categories into the goal-image embedding space learned through `ImageNav` training. This mapping is learned using an in-domain dataset containing 14K images with object category labels.

– **CLIP on Wheels (CoW)** [21]: builds an occupancy map by projecting depth observations, then searches the environment with frontier-based exploration [36]. At each step, CoW calculates a 3D saliency map using a depth and RGB observations and the goal object category via Grad-CAM [26], a gradient-based visualization technique. When the 3D saliency exceeds a threshold the agent navigates to that location and stops. As such, CoW does not require a learned navigation policy.

**Fully-Supervised ObjectNav.** To understand the gap to fully-supervised `ObjectNav` methods, we compare with OVRL [16], a two-stage framework that achieves state-of-the-art `ObjectNav` results in our single RGB camera setting. We highlight OVRL in blue to indicate the use of direct supervision.

## 5.2 Zero-Shot Object-Goal Navigation

In Table 1 we report zero-shot `ObjectNav` performance. We compare with ZER [18] in Table 1a using agent configuration `A`. Notice that our agent is stronger than ZER on `ImageNav`, which is the base pretraining task before ObjectNav can be studied. Specifically, we observe a 7.7% improvement in `ImageNav` `SR` (29.2% → 36.9%). This improvement results from (1) learning to navigate to semantic goal embeddings (as proposed in this work) instead of navigating to image-goal embeddings that are learned from scratch (as done in ZER), (2) using more diverse training environments, and (3) from using a pretrained visual encoder. We provide additional comparisons with ZER using the same set of training environments and without using visual encoder pretraining in Section 5.3, where we also observe improved performance. In Table 1a, we see even larger improvements in `ObjectNav` `SR` of 20.0% (11.3% → 31.3%). These results indicate that our design decisions are particularly useful for zero-shot `ObjectNav`.

In Table 1b we compare with CoW [21] using agent configuration `B`. In `ObjectNav` on the MP3D validation set, we find that training a `SemanticNav` agent improves `ObjectNav` `SR` by 4.2% absolute and 37.8% relative (11.1% → 15.3%). These results demonstrate that learning a navigation policy improves zero-shot `ObjectNav` `SR` over the hand-designed exploration strategy and stopping criteria proposed by CoW. Moreover, we expect further improvements in zero-shot `ObjectNav` performance from scaling our approach (e.g., by collecting more training environments). Such scaling is simply not possible with heuristic methods such as CoW because the navigation policy is not learned. The `SPL` of our approach is 1.5% lower than CoW. However, unlike CoW, our agent navigates without depth observations, which may reduce path efficiency. On HM3D we find that our agent achieves a strong `SR` of 25.5% and `SPL` of 12.6%. Impressively, this zero-shot `SPL` matches OVRL [16], which is directly trained on 40k human demonstrations [10] for the `ObjectNav` task with imitation learning.

## 5.3 Comparison with ZER without Encoder Pretraining and Training Environment Diversity

In Table 2, we train our approach in Gibson environments (instead of HM3D) and do not use a pretrained observation encoder. These settings match ZER [18], allowing for a direct comparison between the two methods. We observe that our approach results in a 4.0% absolute and 35% relative improvement in zero-shot ObjectNav success (11.3% → 15.3%). These results demonstrate that learning to navigate to semantic-goal embeddings outperforms the inverse approach proposed by ZER of first training for image-goal navigation, then learning a mapping from object categories into the image-goal embedding space.

Table 2: **Comparison with ZER** [18] using a ResNet-9 and the Gibson dataset with our approach. Learning SemanticNav (Ours) outperforms learning ImageNav then language grounding (ZER [18]).

| Method | Visual Encoder | Training Dataset | ImageNav (Gibson) | | ObjectNav (Gibson) | |
|---|---|---|---|---|---|---|
| | | | SPL | SR | SPL | SR |
| ZER [18] | ResNet-9 | Gibson | 21.6% | 29.2% | - | 11.3% |
| Ours | ResNet-9 | Gibson | **22.8%** | **33.3%** | **7.4%** | **15.3%** |

## 5.4 Additional Ablations

In Table 3, we study the impact of two key design decisions within our method: (1) the visual observation encoder and (2) the number of training environments. While pretraining the visual observation encoder is known to improve visual navigation task performance (demonstrated in [16]), here we study the impacts on zero-shot transfer to ObjectNav. We find that OVRL pretraining improves ImageNav success by 4.5% (rows 1 vs. 3) or 5.8% (rows 2 vs. 4) depending on the dataset used for training. However, the impact on zero-shot ObjectNav performance is substantially larger. Specifically, ObjectNav success improves by 9.4% (rows 1 vs. 3) and 10.4% (rows 2 vs. 4). These results suggest that a strong visual encoder is often essential for zero-shot transfer to ObjectNav.

In rows 3 vs. 4, we switch the training dataset from the 72 Gibson [4] training environments (row 3) to the 800 (unannotated) HM3D [20] training environments. Surprisingly, we observe a 0.9% drop in ImageNav success, yet a 6.6% improvement in ObjectNav success (rows 3 vs. 4). A similar trend is observed in rows 1 vs. 2. These trends indicate that training environment diversity is particularly useful for zero-shot ObjectNav.

Table 3: **Ablations** of the visual encoder and dataset used for training our SemanticNav agents.

| # | Visual Encoder | Training Dataset | ImageNav (Gibson) | | ObjectNav (Gibson) | |
|---|---|---|---|---|---|---|
| | | | SPL | SR | SPL | SR |
| 1 | ResNet-9 from scratch | Gibson | 22.8% | 33.3% | 7.4% | 15.3% |
| 2 | ResNet-9 from scratch | HM3D | 23.4% | 31.1% | 9.5% | 20.9% |
| 3 | OVRL (ResNet-50, pretrained) | Gibson | 27.6% | **37.8%** | 10.0% | 24.7% |
| 4 | OVRL (ResNet-50, pretrained) | HM3D | **28.0%** | 36.9% | **12.0%** | **31.3%** |

## 5.5 Qualitative Analysis

In Fig. 3, we present qualitative examples of our agent navigating to more complex object descriptions (e.g., *"Find a bathroom sink"*). In each trial, the agent starts at the same position and heading (next to the front door looking into the house). The only thing that changes about the initial conditions is the instructions given to the agent (*"Find a..."* *"...bathroom sink"*, *"...kitchen sink"*, *"...sink and a toilet"*, or *"...sink and a stove"*). Since the agent's policy is stochastic, we show 5 sampled rollouts and highlight the first run in bold colors.

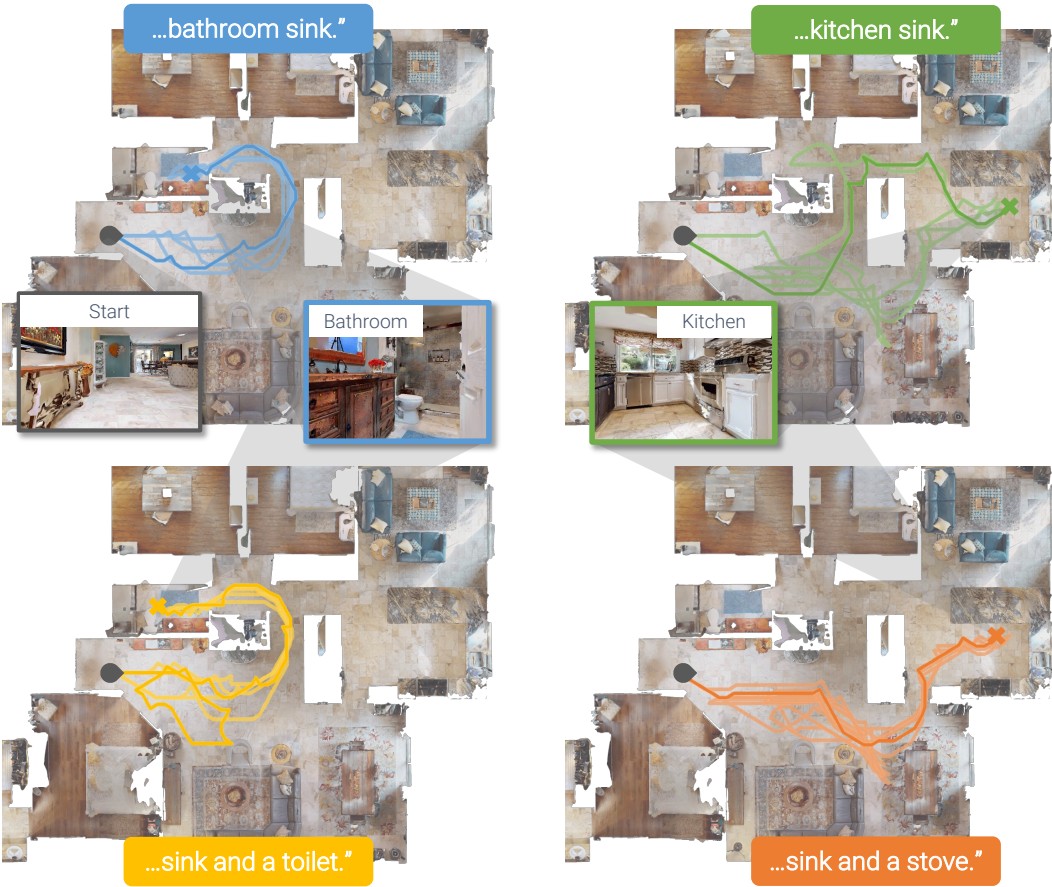

Figure 3: **Qualitative examples** for navigating to complex object descriptions. For each trail, the agent is spawned at the *start* position looking into the house (i.e., to the right on the maps) and given one of four instructions. Each instruction is run five times with the path for the first trail highlighted in bold colors. Our agent appropriately navigates to the correct rooms, demonstrating an understanding of both explicit (*"Find a kitchen sink"*) and implicit (*"Find a sink and a stove"*) room information.

We find that given room information such as *"bathroom"* or *"kitchen"*, the agent appropriately finds a *"sink"* in the corresponding rooms in the house. Furthermore, in these examples the agent does not enter the *"kitchen"* when prompted to look for a *"bathroom sink,"* and vice-versa. In these long trajectories (ranging from 88 to 225 steps), we observe more exploration in the living room and direct navigation when target rooms are visible. We qualitatively observe interesting learned behaviors – for instance, the agent often performs a 360° turn before navigating, possibly to survey the environment.

Next, we experiment with variations in which room information can be inferred from the instruction, but is not explicit. We use *"sink and a toilet"* to indicate *"bathroom"* and *"sink and a stove"* for *"kitchen"*. In these examples, we discover that our agent still navigates to the correct rooms, suggesting that it learns some priors of indoor spaces, such as that a *"stove"* is often found within a *"kitchen."*

## 6 Discussion

We present a *zero-shot* method for learning *open-world* object-goal navigation (`ObjectNav`). Our approach involves projecting image-goals into a semantic-goal embedding space using an image-and-text alignment model (CLIP). This creates a semantic-goal navigation task that does not require annotated

3D environments or collecting human demonstrations. Thus, our method is easy to scale. We discover that `SemanticNav` agents outperform previous zero-shot `ObjectNav` methods, and we identify two factors that have a strong impact on navigation success – pretraining the visual encoder and training in a diverse set of environments. In an open-world setting, we observe navigation patterns that suggest that `SemanticNav` agents can understand complex instructions, such as *"Find a sink and a stove."*

**Limitations and Impact.** `SemanticNav` agents appear to learn useful priors of indoor environments such as which room contains a *"stove."* However, agents may struggle in scenes where a navigation target is in an unusual location (e.g., a stove in a bedroom). Biases in the 3D environments used to train such agents might exaggerate these issues and affect deployments in non-traditional settings. Thus, interventions to mitigate this problem should be considered. Future work might explore how to use the natural language interface to `SemanticNav` agents to guide exploration in such scenarios.

## Acknowledgements

The Georgia Tech effort was supported in part by NSF, ONR YIP, ARO PECASE, and ARL. The views and conclusions contained herein are those of the authors and should not be interpreted as necessarily representing the official policies or endorsements, either expressed or implied, of the U.S. Government, or any sponsor.

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
