# OpenReview forum: "ZSON: Zero-Shot Object-Goal Navigation using Multimodal Goal Embeddings"
_NeurIPS.cc/2022/Conference — NeurIPS 2022 Accept_

### Official Review · Reviewer_mStZ · 2022-06-30

**Rating:** 5
**Confidence:** 5
**Soundness:** 2 fair
**Presentation:** 3 good
**Contribution:** 2 fair

**Summary:**

The problem of zero-shot object navigation has been addressed in the paper. The idea is to project images into a joint space of images and text using a CLIP encoder. Then, the agent is trained to navigate towards images sampled from the environments (which does not require any annotation). At test time, the target categories are specified using text embedded in that joint space.

**Questions:**

- L151: "substantially improves training time (by ∼3.5x)" --> What is the baseline for this comparison?

- L188 mentions 3000 episodes for 3 levels of difficulty. L190 mentions 9M episodes. It seems they are not consistent.

- It is well known that CLIP text and image embeddings are quite different. Is there any additional trick applied to these embeddings to make them similar?

**Limitations:**

The limitations of the work are discussed, but not the societal impact.


**Strengths And Weaknesses:**

**Pros**

+ The problem that this paper investigates i.e. open-world object navigation is quite important.

+ The paper provides strong zero-shot results.

**Cons**

- The idea is a small increment over the recent works such as [19] and [21]. The paper argues that [19] requires annotation during training (L85). However, their annotations are obtained automatically. So it is not different from the proposed approach in terms of annotation.

- The comparisons with the baseline [20] is not fair. The baseline uses ResNet-9 while the proposed method uses ResNet-50. So it is not clear if the improvement comes from the higher capacity of the model or this particular type of training.

- It is not clear if scaling up can improve this approach (as claimed in L255). Mapping language to vision might not be simply improved by scale since the CLIP encoders are fixed.

- Only a few qualitative examples are shown for more complex instructions. It is not clear if those are cherry-picked examples. So, quantitative analysis should be provided for that claim.

---

> ### Author Response · Authors · 2022-08-02
> **Response to Reviewer mStZ (1/2)**
>
> Thank you for the review. We are encouraged that you find the open-world ObjectNav task addressed in this paper “quite important.” We address your concerns below.
>
> > In [19] annotations are obtained automatically.
>
> Not exactly. The method in [19] is directly supervised with ObjectNav annotations. Where do these annotations come from?
>
> In photo-realistic environments (e.g., MP3D), this requires manually labeling objects. In synthetic environments (e.g., RoboTHOR), the objects are automatically annotated. However, transferring from synthetic to photo-realistic environments introduces an additional sim2sim challenge.
>
> Recent work (released in June, after NeurIPS deadline) [ProcTHOR] quantitatively demonstrates this challenge. ProcTHOR trained the agent from [19] in 10,000 synthetic environments. The zero-shot transfer of this agent to HM3D was limited (13.2% success on HM3D-test). For comparison, our approach has a 25.5% success rate in HM3D-val. Notice that to achieve stronger success rates on HM3D, ProcTHOR also needed to train on human annotations in HM3D. Thus, the claim that ObjectNav performance on photo-realistic environments can be achieved with automatically generated annotations is not accurate.
>
> Furthermore, we believe that our approach and [19] could be combined. Specifically, we expect that jointly training for ImageNav (as we propose) and ObjectNav (as in [19]) in a multi-task framework would lead to further improvements. Thus, we view these lines of research as complementary.
>
> > Small increment over [21].
>
> We disagree.
>
> As already discussed in L99-103, [21] is concurrent work that combines a heuristic, rule-based navigation policy with CLIP for ObjectNav. We propose a new method for learning a navigation policy that substantially outperforms [21] and opens the potential for further scaling (e.g., by collecting more unannotated training environments).
>
> Specifically, our approach outperforms [21] by 4.2% in terms of MP3D ObjectNav success rate (Table 1). To contextualize this improvement, consider that between the Habitat 2020 and 2021 ObjectNav challenges, the success rate of the challenge winners improved 5% (25% → 30%). Thus, 4.2% is a substantial improvement over [21].
>
> Furthermore, results in Table 2 suggest that scaling the number of training environments may further improve the performance of our approach. While, it is unclear how to scale the approach in [21].
>
> > “The baseline [20] uses a ResNet-9 while the proposed method uses ResNet-50”.
>
> Yes, and we discuss this difference in L277-278 and ablate this difference in Table 3.
>
> Specifically, we modify our approach to use a ResNet-9 (from scratch) and use the same training environments (Gibson) that were used in [20]. In Table 3, we find that with these settings our approach outperforms [20] by 4% on Gibson ObjectNav SR (11.3% vs. 15.3%).
>
> > It is not clear if scaling up can improve this approach (as claimed in L255).
>
> Table 2 provides direct evidence to support the claim made in L255.
>
> Specifically, switching from 72 Gibson environments to 800 (not 1000) HM3D environments improves success on ObjectNav in Gibson by 6.6% (24.7% to 31.3%). So, we expect that “collecting more training environments” will lead to further improvements (as claimed in L255).
>
> > “Only a few qualitative examples are shown for more complex instructions.”
>
> We provide additional qualitative examples in the appendix to highlight common failure modes of our approach. That said, we agree that quantitative evaluations are ideal, but creating a dataset of complex compositional language queries for ObjectNav is an open research question in itself.
>
> > L151: "substantially improves training time (by ∼3.5x)" --> What is the baseline for this comparison?
>
> The baseline here is encoding goal images with a ResNet-50 (CLIP_v) on-the-fly during training. We improve training time by caching these goal embeddings, which is possible because we freeze the goal encoder (CLIP_v).
>
> > L188 mentions 3000 episodes for 3 levels of difficulty. L190 mentions 9M episodes. It seems they are not consistent.
>
> Nice catch. It should say 3,000 episodes per (not for) difficulty level (easy, medium, and hard); 9,000 episodes per scene; and 7.2M episodes total. Note: we train in 800 (not 1000) HM3D environments. We will revise the wording.
>
> References:
>
> - [ProcTHOR] “ProcTHOR: Large-Scale Embodied AI Using Procedural Generation,” Matt Deitke, Eli VanderBilt, Alvaro Herrasti, Luca Weihs, Jordi Salvador, Kiana Ehsani, Winson Han, Eric Kolve, Ali Farhadi, Aniruddha Kembhavi, Roozbeh Mottaghi, arxiv, 2022

---

> > ### Author Response · Authors · 2022-08-02
> > **Response to Reviewer mStZ (2/2)**
> >
> > > "It is well known that CLIP text and image embeddings are quite different. Is there any additional trick applied to these embeddings to make them similar?"
> >
> > Great question! Yes, there are two specific design choices we found to be useful.
> >
> > During ImageNav training our agent learns to stop near the goal. This means it stops at viewpoints that are visually similar, but not exactly the same as the goal image. Learning this approximation in training transfers to the downstream ObjectNav task.
> >
> > Conceptually, proximity in physical space translates to similarity in the CLIP embedding space. Thus, the agent learns to stop close to goal embeddings (image or text), but does not require an exact match. Exact matching would be problematic, because as the reviewer points out, CLIP text and image embeddings are different.
> >
> > Two design choices (or “tricks”) discourage exact matching. First, we apply image augmentations to agent observation during training (L159-160). Thus, even if the agent navigates to the exact goal location it will never see the exact goal image, so it cannot learn exact matching. Second, we never process agent observations with the CLIP visual encoder -- which would make it easier to learn exact (as opposed to approximate) matching.
> >
> > Beyond these design choices, no additional “tricks” were required. We will add this discussion to the paper.

---

> > > ### Comment · Reviewer_mStZ · 2022-08-09
> > > **final score**
> > >
> > > The rebuttal clarified a few points such as ResNet-9 vs ResNet-50 experiments, tricks used for aligning CLIP text and visual encoders, and inconsistency of the number of episodes.
> > >
> > > Claim in L255: My point was that the zero-shot capabilities of the model will not change by scale since both the visual and textual encoders of CLIP are frozen. Table 2 shows the navigation capability.
> > >
> > > There are still some issues that make the paper weak. E.g., (1) The idea of the paper is incremental compared to [19], (2) Presenting only qualitative examples is not sufficient. Those types of results should be removed or presented quantitatively.
> > >
> > > I am going to increase the rating to Borderline Accept since I still consider this as a borderline paper.

---

> > > > ### Author Response · Authors · 2022-08-09
> > > > **Response to Reviewer mStZ**
> > > >
> > > > We are happy we resolved several of your concerns. Thank you for raising your score!
> > > >
> > > > > Claim in L255: My point was that the zero-shot capabilities of the model will not change by scale since both the visual and textual encoders of CLIP are frozen. Table 2 shows the navigation capability.
> > > >
> > > > For clarity, Table 2 shows *zero-shot* navigation performance (the focus of this work) improves with scale. We do freeze CLIP (and do not claim otherwise).
> > > >
> > > > > The idea of the paper is incremental compared to [19]
> > > >
> > > > We respectfully disagree. [19] uses a CLIP encoder within a traditional ObjectNav training pipeline. We propose a new *zero-shot* training pipeline (leveraging CLIP).
> > > >
> > > > > Presenting only qualitative examples is not sufficient.
> > > >
> > > > We respectfully disagree. But more importantly, this is a stylistic preference (so reasonable people can disagree on this issue) and not a shortcoming of the work.

---

### Official Review · Reviewer_J2xC · 2022-07-11

**Rating:** 5
**Confidence:** 3
**Soundness:** 3 good
**Presentation:** 3 good
**Contribution:** 3 good

**Summary:**

This work proposed the semantic-goal navigation task on the object goal navigation without the provided object-goal annotation. Instead of the conventional object-goal annotation, they converted the image-goal navigation task into semantic-goal navigation (SemanticNav) by aligning goal images from image-goal nav into a multimodal, semantic embedding space for object goal nav.

The key idea is that they use the CLIP representation to transform image-goals and object-goals into semantic-goals representing navigation targets. They trained a semantic-goal navigation agent using image-goals transformed by CLIP. This allows the agent to search for the goal position from both visual and textual supervision.

They compared their model of ZSON with two baselines for SemanticNav: Zero Experience Required (ZER) of mapping a goal object category into goal-image embedding space and CLIP on Wheels (CoW), which is free from the learned navigation policy. They also compared to the conventional supervision model of OVRL. Their experimental results surprisingly suggest that ZSON is competitive or better than the supervised model of OVRL in SPL.

Although their approach for SemanticNav is quite interesting, their approach reminds me of the data augmentation or multi-tasking for the conventional object-navigation task from a different task of the image-goal navigation. If the proposed model is effective, it will surely succeed in the conventional object-goal navigation settings compared to the existing object-goal models. It might be also interesting to compare the multi-tasked models with the state-of-the-art (supervised) object navigation models.

**Questions:**

- If my understanding is correct, it is possible to use your navigation dataset combined with the original object-goal navigation. As you didn’t use the object-goal annotations, you can finetune the semantic-nav agent in the conventional object-goal navigation. Did you experiment with your model in this setting?

**Limitations:**

- Lack of the comparisons with other state-of-the art models in the conventional object-goal navigation setting

**Strengths And Weaknesses:**

**Stregness**
- Proposal to a new task of SemanticNav driven by CLIP joint representation of the target object image and language. This allows the vocabulary target supervision.

**Weakness**
- The proposed model is evaluated in the limited condition. It seems that it is possible to apply the proposed model for the conventional (supervised) object-goal navigation
- Model comparisons are limited. I'd like to see more strong baselines in both the SemanticNav setting and the (supervised) object-goal navigation setting.

---

> ### Author Response · Authors · 2022-08-02
> **Response to Reviewer J2xC**
>
> Thank you for the encouraging feedback. We address your concerns below.
>
> > “It seems that it is possible to apply the proposed model for the conventional (supervised) object-goal navigation.”
>
> Yes, we agree! To clarify, we DO evaluate in the standard object-goal navigation setting -- i.e., same agent, same scenes, same episodes, etc.
>
> > I'd like to see more strong baselines in both the SemanticNav setting and the (supervised) object-goal navigation setting.
>
> Please see above. We already evaluate in the standard supervised ObjectNav setting and do already compare with a supervised ObjectNav baseline in the paper.
>
> Specifically, we provide comparisons with the two existing zero-shot ObjectNav methods (ZER [20] and CoW [21]) and a state-of-the-art supervised ObjectNav method (OVRL [12]) in Table 1. There is a gap to the fully supervised SoTA, which is understandable because this method is supervised with 40k ObjectNav demonstrations that we don’t require with our approach.
>
> > “If my understanding is correct, it is possible to use your navigation dataset combined with the original object-goal navigation. As you didn’t use the object-goal annotations, you can finetune the semantic-nav agent in the conventional object-goal navigation. Did you experiment with your model in this setting?”
>
> Great question! We followed the reviewer’s suggestion and finetuned our agent with ObjectNav annotations via RL. We see 7.6% - 24.1% absolute improvements in ObjectNav success rate (SR).
>
> Specifically, we initialized from our zero-shot (semantic-nav) agent that was trained for 500M steps of experience on HM3D ImageNav episodes (Table 1b).
>
> In finetuning experiment #1, we finetune for only 25M steps on MP3D ObjectNav episodes using a similar RL approach as described in the paper. With this finetuning, MP3D ObjectNav SR improves by 7.6% absolute (15.3% → 22.9%) and MP3D ObjectNav SPL improves by 4.4% absolute (4.8% → 9.2%). This SPL of 9.2% surpasses the SoTA in the RGB-only setting, which is 7.0% SPL and was achieved by a directly supervised ObjectNav method, OVRL [12] (Table 1).
>
> In finetuning experiment #2, we finetune for 100M steps on HM3D ObjectNav episodes via RL. With this finetuning, HM3D ObjectNav SR improves by 24.1% absolute (25.5% → 49.6%) and HM3D ObjectNav SPL improves by 14.4% absolute (12.6% → 27.0%). These results exceed the OVRL [12] supervised learning baseline from Table 1 (which was trained on 40k human demonstrations in MP3D environments) by 16.8% absolute on HM3D ObjectNav SR (32.8% vs. 49.6%) and 14.7% absolute on HM3D ObjectNav SPL (12.3% vs. 27.0%).
>
> Thank you again for this suggestion!

---

> > ### Author Response · Authors · 2022-08-10
> > **Response to Reviewer J2xC**
> >
> > Thank you again for your review and your suggestions! Do you have any remaining questions? If not, please consider updating your recommendation in light of the clarifications.

---

> > ### Comment · Reviewer_J2xC · 2022-08-10
> > **final score**
> >
> > Thank you for your replies,
> >
> > My concerns are almost resolved with demonstrated interesting experimental results. Will you update the manuscript and include these results of the semantic-nav agent in the conventional object-goal navigation (#1 and #2 experiments)? Or they are already in the manuscript? If you include them in the manuscript, I'm going to update my score to WA.

---

> > > ### Author Response · Authors · 2022-08-10
> > > **Response to Reviewer J2xC**
> > >
> > > Yes, we will include those experimental results in the paper.

---

### Official Review · Reviewer_tra9 · 2022-07-12

**Rating:** 4
**Confidence:** 5
**Soundness:** 2 fair
**Presentation:** 3 good
**Contribution:** 1 poor

**Summary:**

The paper addresses ImageNav and ObjectNav tasks, using pre-trained CLIP representations.

**Questions:**

See above

**Limitations:**

Transfer of existing generalised representations is only useful if those representations can be suitably adapted to specific downstream tasks (maximising performance of the approaches therein), while retaining their cross-task generalisability. I feel like this discussion is important, but it is missing from the present manuscript.

**Strengths And Weaknesses:**

Strengths:

The paper is well-written.

I do appreciate the discussion on open-world scenarios.

Weaknesses:

Section 1: The manuscript states "An important advantage of our approach is that it reduces the data labeling burden."  To be precise, the approach proposed in the manuscript, itself, does not reduce the data-labeling burden; instead, it enjoys the reduction in required data-labeling through the use of CLIP — particularly since CLIP was not even fine-tuned on the task.

Section 1: I remain somewhat unconvinced by the approach proposed by the manuscript. Arguably, the approach *has* observed general information about the task, through CLIP’s (pre-)training step and that the proposed approach is just utilising this background experience to perform somewhat above-random on the transfer task.

Section 4 / Figure 2: I have a strong concern with the novelty claims in this manuscript. CLIP is used off-the-shelf (with no effort to adapt CLIP to the tasks), visual encoder is pre-trained, policy network architecture and training paradigm is taken from the plethora of related approaches in embodied tasks.

Section 5 / Table 1: Experiments are incomplete? Partial experiments and different baselines on different tasks? Some consistency is required.

---

> ### Author Response · Authors · 2022-08-02
> **Response to Reviewer tra9**
>
> Thank you for the feedback. We are glad that you found the paper “well-written” and appreciated the “the discussion on open-world scenarios”. We address your comments below.
>
> > The proposed approach “does not reduce the data-labeling burden.”
>
> The reviewer appears to be conflating two different data-labeling burdens. Our approach DOES NOT reduce the data-labeling burden for training CLIP (and never claims to). Our approach DOES unambiguously reduce the data-labeling burden for ObjectNav (which is what we claim).
>
> As described in L50-52: ObjectNav training requires ObjectNav annotations. One needs to either (a) label objects in 3D environments to determine if an agent reaches a goal during training or (b) collect human demonstrations, as done in [8]. Zero-shot ObjectNav methods (such as the one that we propose) do not require either form of manual annotation (for ObjectNav).
>
> Specifically, we sidestep this data-labeling requirement by training for ImageNav, which does not require labeled environments or human demonstrations. And then, we enable zero-shot transfer to ObjectNav with CLIP. Simply the existence of an annotated dataset for training CLIP does not automatically provide ObjectNav annotations; our approach overcame that gap.
>
> > The proposed approach uses CLIP’s “background experience to perform somewhat above-random on the transfer task.”
>
> This mischaracterizes the results. We are unsure what result backs up the reviewer’s claim of “somewhat above-random.”
>
> In ObjectNav, random agents succeed 0% of the time (see the last entry on the Habitat 2020 ObjectNav Challenge Leaderboard -- link below). Random agents never reach the goal.
>
> In contrast, as stated on L57-60 and found in Table 1, on the Gibson dataset our agent achieves 31.3% success rate, which is a 20.0% absolute improvement over previous zero-shot ObjectNav results. In MP3D, our agent achieves 15.3% success rate, a 4.2% absolute gain over existing zero-shot methods.
>
> If the reviewer provides a basis for the “somewhat above-random” performance claim, we can try to further address this concern.
>
> > Novelty.
>
> As described in L45-48, the novelty in our approach is using ImageNav to train agents that find latent representations produced by CLIP, which facilitates zero-shot transfer to ObjectNav. To the best of our knowledge, this is the first approach to propose such a solution. If the reviewer can point to prior work that has proposed a similar solution to ours, we will be happy to address it.
>
> > “Table 1: Experiments are incomplete?”
>
> No. Our experiments are fairly exhaustive and extensive. To quote EtZg, “The evaluations are done on several different datasets and show consistent improvement over zero-shot baseline” and mStZ “The paper provides strong zero-shot results.” Prior work is missing results on certain datasets, which is why they appear blank in our tables.
>
> Specifically, in Table 1 we report results on 3 ObjectNav datasets (Gibson, HM3D, and MP3D). As described on L205-210, this required training agents in two different configurations. This collection of datasets allowed directly comparing with the two existing methods for zero-shot ObjectNav [20, 21]. Unfortunately, prior work [20, 21] did not report results on all three dataset (or did not report SPL), so those entries in Table 1 are blank.
>
> References:
>
> - Habitat 2020 ObjectNav Challenge Leaderboard: https://eval.ai/web/challenges/challenge-page/580/leaderboard/1634

---

> > ### Author Response · Authors · 2022-08-09
> > **Response to Reviewer tra9**
> >
> > Thank you again for your review. Do you have any remaining questions? If not, please consider updating your recommendation in light of the clarifications.

---

> > ### Comment · Reviewer_tra9 · 2022-08-09
> > **Final score**
> >
> > I would like to thank the authors for clarifying several points.
> >
> > A couple outstanding questions/concerns remain:
> >
> > > Re: The proposed approach “does not reduce the data-labeling burden.”
> >
> > In the context of transferring representations, the stated data-labeling burdens are one and the same. One cannot claim that an approach reduces the data-labeling burden of a task, if it relies on an off-the-shelf, sample-inefficient, large-capacity model — even if that model was pre-trained on a different task from the deployment task. The pre-trained model reduced the data-labeling burden, with its pre-trained representations available for transfer. It is difficult to say that an approach that uses said representations (even for a different task) reduces the task's data-labeling burden. For this reason, I still think that using this as a primary contribution is inappropriate.
> >
> > > Re: “Table 1: Experiments are incomplete?”
> >
> > Since those prior works do not report results on those datasets, how can their baselines serve as strong comparisons for contextualising the performance of the proposed method? With partial experiments and different baselines on different tasks, a reader may find it difficult to glean substantial insights from Table 1 for this reason.
> >
> > I am willing to increase my score to Borderline Reject, as I still regard this as a borderline paper.

---

> > > ### Author Response · Authors · 2022-08-10
> > > **Response to Reviewer tra9**
> > >
> > > Thank you for raising your score! We address your two remaining concerns below.
> > >
> > > > It is difficult to say that an approach that uses [pre-trained] representations (even for a different task) reduces the task's data-labeling burden.
> > >
> > > No, it is not difficult at all. In fact, that is the very reason for pursuing zero-shot methods.
> > >
> > > More importantly, vanilla transfer learning requires labeled data for the downstream task. Before our approach, such methods for ObjectNav trained with tens of thousands of 3D semantic category annotations. Our approach shows how to use 0. In what sense is that not a reduction?
> > >
> > > > Since those prior works do not report results on those datasets, how can their baselines serve as strong comparisons for contextualising the performance of the proposed method?
> > >
> > > A reader can directly compare our method with each baseline on the datasets reported by each baseline. A reader cannot directly compare the baselines to each other, which is a limitation of the experimental evaluation in prior work, not in our approach.

---

### Official Review · Reviewer_Et7g · 2022-07-15

**Rating:** 8
**Confidence:** 4
**Soundness:** 4 excellent
**Presentation:** 4 excellent
**Contribution:** 4 excellent

**Summary:**

The paper proposes a technique that enables zero-shot object-goal navigation: given a text description of an object that is reachable in a 3D environment, an agent needs to find the object using visual input. The contribution of the paper mainly lies in that it utilizes a pre-trained visual-language model (i.e. CLIP) to project the image goal of the object and the textual description of the object into the same embedding space, such that a navigation agent trained on only image goals can perform zero-shot navigation when only given textual goal at test time. The proposed method is evaluated on three datasets and is shown to outperform prior methods in zero-shot setting. Because CLIP is pretrained on large-scale image-language pairs, this approach also enables the possibility of specifying navigation goal in free-form language at test time, such as “find a kitchen sink”.

**Questions:**

Please refer to the comments in the weakness section.

To summarize and re-iterate:

1. In Table 2, even though the dataset is scaled to 1000 envs from 72 envs, why is there decreased performance for ImageNav?
2. And why is there only 6.6% improvement in ObjectNav success rate, given the significant increase of training envs?

**Limitations:**

Yes, the authors have adequately addressed certain limitations of the method.

**Strengths And Weaknesses:**

Strengths:

1. The paper text is written with clarity and figures are well made.
2. The method appears to be novel for the reviewer.
3. The evaluations are done on several different datasets and show consistent improvement over zero-shot baseline, though there still exists certain gap to supervised baseline.
4. The most significant strength of the paper is that it investigates an important question in the field: how to leverage internet-scale data (or existing large models pre-trained on these data) for open-world embodied tasks? This is particularly important because any dataset from a single (or a few) domains would not be able to capture the complexity of the open world, so leveraging large-scale unstructured data is one of the few promising approach to achieve this. This paper investigates exactly this and shows a meaningful step forward.

Weaknesses:

1. Though requiring unannotated data, the ImageNav agent presented in this paper relies on large amount of data for training. And even by significantly increasing the dataset size, as shown in Table 2, only moderate improvement is shown on ObjectNav (along with decreased performance in ImageNav). It raises a question about how scalable the ImageNav agent is, i.e. how much more improvement in success rate can one expect by continuing scaling the training data.

---

> ### Author Response · Authors · 2022-08-02
> **Response to Reviewer Et7g**
>
> Thank you for the encouraging review. We are happy you found that our paper “investigates an important question in the field” and provides “a meaningful step forward.” We address your questions below.
>
> > “In Table 2, even though the dataset is scaled to 1000 envs from 72 envs, why is there decreased performance for ImageNav?”
>
> Good question.
>
> Our current hypothesis is that this is due to the choice of the learning rate and the scale of training. Specifically, both HM3D and Gibson ImageNav results in Table 2 are reported for 500M steps of training. However, results in [12] indicate that long-training schedules are helpful for HM3D -- success rates do not saturate through 2 Billion steps. Gibson is a smaller dataset and success rises and satures much quicker. So, in Table 2 the reason it appears that a larger dataset does not help for ImageNav is only because we compare results at 500M steps.
>
> Interestingly, for zero-shot ObjectNav the trend is reversed: HM3D outperforms Gibson (Table 2). As discussed in L274-275, these trends suggest that increasing environment diversity is particularly helpful for zero-shot transfer to ObjectNav.
>
> > “And why is there only 6.6% improvement in ObjectNav success rate, given the significant increase of training envs?”
>
> To contextualize this improvement, consider that between the Habitat 2020 and 2021 ObjectNav challenges, the success rate of the challenge winners only improved 5% (from 25% to 30%). In other words, SoTA on this task advances slowly. So, a 6.6% improvement is an encouraging signal.

---

### Meta-Review · Area_Chair_xouH · 2022-08-25

**Recommendation:** Accept
**Confidence:** Less certain

**Metareview:**

This work introduced a method for object-goal navigation in open-world settings. The crux of the approach is to use a pre-trained vision-and-language model CLIP to associate embeddings of image goals and textual descriptions of the objects such that the model can perform zero-shot navigation given textual goals.

This paper received mixed reviews from four expert reviewers, ranging from Borderline Reject to Strong Accept. The reviewers appreciated the problem motivation, the novel method of using large-scale pre-trained models for embodied AI, and the demonstrated zero-shot navigation performances. Nonetheless, concerns have been raised by the reviewers, including the limited evaluations and unfair baseline comparisons.

The authors did a good job easing many of the concerns raised in the initial reviews. They successfully swayed Reviewer mStZ to update their rating to Borderline Accept and Reviewer tra9 to Borderline Reject. In addition, Reviewer J2xC mentioned that they will update their rating to Weak Accept if the authors include the results in the new revision, which the authors promised to do. The only reviewer who held a negative opinion after the discussion period is Reviewer tra9, who argued against the authors' claim that their approach reduces the data labeling burden (by using pre-trained CLIP) and pointed out that different baselines are used for different tasks. The AC read the paper, the reviews, and the reviewer-author discussions carefully and found that the authors made convincing arguments in response to Reviewer tra9's criticisms. Given that the majority of the reviewers held positive opinions on this work and no major flaw had been identified, the AC would recommend accepting this work at NeurIPS and letting the community judge the merit of this work. The AC would strongly suggest the authors incorporate all reviewers' comments in the final version.

**Award:**

No

---

### Decision · Program_Chairs · 2022-09-14

Accept